# Space-time clustering and temporal trend analysis of pulmonary tuberculosis in Brazil, 2010–2023

José Mário Nunes da Silva[1,2]*, Fredi Alexander Diaz-Quijano[2], Lívia Teixeira de Souza Maia[3], Mauro Niskier Sanchez[1,4], Ximena Pamela Claudia Díaz Bermúdez[4], Eduardo de Souza Alves[5], Walter Massa Ramalho[1,4]

**1** Center for Tropical Medicine, School of Medicine, University of Brasília, Brasília, Federal District, Brazil, **2** Department of Epidemiology – Laboratório de Inferência Causal em Epidemiologia (LINCE-USP), School of Public Health, University of São Paulo, São Paulo, São Paulo, Brazil, **3** Center for Public Health, Federal University of Pernambuco, Vitória de Santo Antão, Pernambuco, Brazil, **4** Department of Public Health, School of Health Sciences, University of Brasília, Brasília, Federal District, Brazil, **5** Ministry of Health, Brasília, Federal District, Brazil

* zemariu@hotmail.com

## Abstract

### Background

Identifying high-risk areas for pulmonary tuberculosis (PTB) is essential for understanding the dynamics of disease transmission and for guiding more effective control strategies. Therefore, this study aimed to analyze the temporal trends and spatiotemporal distribution of PTB across Brazilian municipalities between 2010 and 2023.

### Methods

This is an ecological study using data on PTB cases reported in Brazil's Notifiable Diseases Information System (SINAN) from 2010 to 2023. Time series analysis, global and local spatial autocorrelation, and space-time scan techniques were applied to identify temporal trends and spatial patterns of the disease at the municipal level.

### Results

A total of 896,062 new PTB cases were analyzed. Notification peaked mainly in March and August. The average incidence rate was 30.3 cases per 100,000 inhabitants-years. An increasing trend was observed in 13 states and the Federal District. Spatial analysis identified 804 municipalities as hotspots, with 168 showing persistent high incidence throughout the study period. A total of 28 spatiotemporal clusters were detected, involving 379 municipalities, of which 212 were classified as high priority. The most likely cluster was located in the Rio de Janeiro Metropolitan Region, encompassing 10 municipalities, with a relative risk (RR) of 2.51 between

**Data availability statement:** All relevant data are within the manuscript and its Supporting information files.

**Funding:** This study was funded by Brazilian National Council for Scientific and Technological Development (CNPq) – grant number 444969/2023-3. The funders had no role in study design, data collection and analysis, decision to publish, or preparation of the manuscript.

**Competing interests:** The authors have declared that no competing interests exist.

2017 and 2023. Spatial variation in temporal trends identified 22 additional clusters, including a prominent cluster composed of 493 municipalities in the Legal Amazon, which showed an internal time trend of 1.99% annual growth and a RR of 1.51.

## Conclusion

The study identified persistent and expanding patterns of PTB in Brazil at both regional and national levels, revealing specific areas with higher burden and increasing trends that should be prioritized. These findings provide evidence to support decision-making at federal level while reinforce the need for regionally tailored surveillance and control strategies to ensure a more effective and equitable response to tuberculosis across the country.

---

## Introduction

Tuberculosis (TB) has once again become the leading cause of death from a single infectious agent worldwide, accounting for an estimated 1.25 million deaths in 2023 [1]. In the same year, Brazil reported 84,994 new TB cases, and TB-related mortality, after declining up to 2020, resumed an upward trajectory, reaching 6,025 deaths [2]. Despite the progress achieved in recent decades, global efforts remain insufficient to meet the goal of eliminating TB as a public health problem by 2035 [3].

As an airborne infectious disease, pulmonary tuberculosis (PTB) has a high potential for dissemination, affecting not only areas with high incidence but also adjacent regions [4]. In a country as large and diverse as Brazil, TB distribution is highly heterogeneous, reflecting deep social and structural inequalities. The persistence of poverty, housing shortages, overcrowded environments, malnutrition, and limited access to health services contributes to maintaining endemic transmission, particularly in urban peripheries and territories with poor living conditions [5–7]. Incarceration, migration, and informal labor further exacerbate vulnerability, while regional disparities in health-system capacity and social protection reinforce unequal exposure and outcomes [8,9].

These inequalities explain why certain municipalities and regions, such as large metropolitan areas, border zones, and parts of the Legal Amazon, have remained persistent TB hotspots over time [5,6,10,11]. Identifying these spatial disparities is therefore crucial for advancing toward equitable TB control and aligning public health strategies with the broader social determinants of health.

In this context, spatial and spatiotemporal cluster analyses are well-suited to this task because they detect areas with more cases than expected and reveal patterns of persistence and spread [6,12]. Moreover, examining regional variations in disease trends allows for the evaluation of current interventions and helps identify where prevention and control strategies have yet to achieve the desired impact [5,7]. From a health management perspective, this approach supports the updating and adaptation of public health policies and enhances the allocation of resources to areas of highest priority [4,7,13]. Thus, this study aimed to analyze the temporal trends and spatiotemporal distribution of PTB in Brazilian municipalities between 2010 and 2023.

## Methods

### Study design and setting

This is an ecological study with mixed design, using Brazil's 5,570 municipalities as units of analysis. Brazil, located in South America, is divided into five major geographic regions (North, Northeast, Central-West, Southeast, and South), which encompass 26 states and the Federal District [14]. Within this national territory, there is an area known as the Legal Amazon, which comprises 772 municipalities distributed across all the states in the Northern Region (Amazonas, Acre, Rondônia, Roraima, Amapá, Tocantins, and Pará), as well as the state of Mato Grosso (Central-West) and part of Maranhão (Northeast). This areas accounts for approximately 58.9% of the national territory and is home to 77% of the country's Indigenous population (S1 Fig in S1 File) [15].

### Study population and data source

The study encompassed all newly reported cases of PTB between 2010 and 2023 in the Brazil's Notifiable Diseases Information System (SINAN), managed by the Department of Informatics of the Unified Health System (DATASUS) (http://www2.datasus.gov.br). SINAN is the Brazilian system responsible for recording and processing data related to notifiable diseases nationwide, such as TB, and plays a key role in public health surveillance [16].

Census and intercensal population data for 2010–2023 were obtained from DATASUS, sourced from the Brazilian Institute of Geography and Statistics (IBGE) (https://www.gov.br/saude/pt-br/composicao/seidigi/demas/dados-populacionais).

### Variables

The incidence rate of PTB was calculated as the total number of reported cases from 2010 to 2023 divided by the resident population over the same period, expressed per 100,000 inhabitants per year. Records with missing sex (n = 76; 0.008%), age (n = 235; 0.03%), or municipality of residence (n = 168; 0.02%) were handled by proportional redistribution: (i) for sex and age, within the same municipality and calendar year; and (ii) for municipality of residence, within the same state and calendar year. This is equivalent to single imputation using empirical frequencies under missing completely at random (MCAR) or missing at random (MAR) assumptions, preserving totals and avoiding artificial rate deflation [17]. Two caveats apply: it does not propagate imputation uncertainty and could be biased if missingness were missing not at random (MNAR) [18]. However, given the extremely low missingness, uniform patterns across strata, and the data-collection workflow, we find no evidence of an MNAR mechanism [19].

The epidemiological characteristics of the cases included: sex, age (in years), self-reported race/ethnicity, education in years, area of residence, smoking status, alcohol consumption, illicit drug use, presence of comorbidities such as AIDS, diabetes mellitus, and mental disorders, as well as HIV and bacteriological status.

### Statistical analysis

**Time series analysis.** We initially constructed a monthly time series of PTB cases to analyze their behavior and trends over the study period. For this purpose, we applied the Seasonal Trend Decomposition using Loess (STL) method, which is based on locally weighted regression [20]. This analysis was conducted using R software (version 4.4.2, R Core Team, Vienna, Austria), with the support of the forecast package.

In addition, we used the Joinpoint Regression Program (version 5.3.0.0, National Cancer Institute, Bethesda, MD, USA) to assess temporal trends. This model identifies whether a given indicator exhibits a stationary, increasing, or decreasing trend and determines the time points at which changes in these trends (joinpoints) occur. The following parameters were adopted for this analysis: log-linear regression model, minimum of 0 joinpoints, maximum of 3 joinpoints, model selection based on the Monte Carlo permutation test with 4,499 permutations, and first-order autocorrelated errors based on the data [21]. These settings enabled the estimation of the Annual Percent Change (APC) and the Average Annual Percent Change (AAPC), both with 95% confidence intervals (95% CI) and a significance level of 5%.

**Spatial autocorrelation analysis.** We used the Global Moran's Index (Moran's I) to identify spatial autocorrelation and detect the spatial distribution pattern of PTB across Brazilian municipalities [22]. Moran's I values range from −1 to +1: a positive value indicates positive spatial correlation, while a negative value suggests negative correlation, with high and low values interspersed [23]. Values near zero indicate no spatial clustering, suggesting that the data are randomly distributed [23]. To assess the significance of the spatial autocorrelation, we calculated the z-score and p-value using randomization procedures with 999 iterations via Monte Carlo simulations.

Subsequently, we applied the Local Getis-Ord $G_i^*$ statistic to identify local spatial autocorrelation and determine the location of clusters or spatial concentration zones (hotspots and coldspots). This analysis considered a neighborhood matrix based on the average number of events in the six nearest neighboring municipalities [24]. The output of this analysis included a z-score and corresponding p-value for each municipality. High positive z-scores indicated clusters of high values (hotspots), while negative z-scores indicated areas of lower occurrence (coldspots) [24]. A z-score above ±1.96 was interpreted as statistically significant, indicating a $p$-value $< 0.05$. All analyses were also performed using R software, employing spatial autocorrelation packages such as spdep and sfdep.

**Space-time scan statistic analysis.** To identify spatiotemporal clusters of municipalities at higher risk of PTB transmission, we used Kulldorff's space-time scan statistic [25]. This approach is based on a discrete Poisson probability model and a maximum likelihood ratio test [26]. The method involves the use of a moving cylindrical window, where the base represents the geographic area and the height corresponds to the temporal dimension [27]. This process allows for the identification of clusters in specific areas and the assessment, during a given time period, of whether there is a higher or lower proportion of cases compared to other areas under analysis [28].

Additionally, we applied the Spatial Variation in Temporal Trends (SVTT) technique, which differs from the previous methods by evaluating temporal trends within the clusters. The temporal trend is analyzed both inside and outside the scanning circle: the variation in trend within the cluster is referred to as the internal time trend (ITT), while the trend in the surrounding areas is termed the outside time trend (OTT). In this analysis, statistical significance is assessed for the temporal trends (ITT and OTT), rather than for cluster formation as in spatial and space-time scanning [7].

In these analyses, the relative risk (RR) is calculated as the ratio between the estimated risk inside the cluster and the estimated risk outside the cluster [26]:

$$RR = \frac{c/E[c]}{(C-c)/(C-E[c])}$$

Where $c$ is the number of observed cases within the cluster, $C$ is the total number of cases within the scanning window, and $E[c]$ is the expected number of cases, adjusted for covariates, under the null hypothesis.

Similarly, the logarithmic likelihood ratio (LLR) is calculated for various circular window centers and radii by comparing PTB incidence rates inside and outside the window [26]:

$$LLR = \log\left(\frac{c}{E[c]}\right)^c \left(\frac{C-c}{C-E[c]}\right)^{C-c} \cdot I()$$

Where $(C-E[c])$ is the expected number of cases outside the window, and $I()$ is an indicator function.

Thus, the window with the highest LLR value was identified as the most likely cluster, while secondary clusters were defined as other windows with statistically significant LLR values. Statistical significance was evaluated through Monte Carlo simulations (999 replications), and clusters with a $p$-value $< 0.05$ were considered statistically significant [26].

To mitigate statistical noise and enhance the interpretability of results, clusters composed of two or fewer municipalities were excluded from the analysis, as recommended [26]. Very small clusters are more susceptible to random variability and to the modifiable areal unit problem (MAUP), which can lead to unstable incidence estimates and potentially spurious patterns or false positives arising from random fluctuation [5,29]. In this analysis, the maximum radius of the cylinder base was set to encompass up to 50% of the total at-risk population, and the maximum temporal size of clusters was limited to 50% of the study period. Age group and sex of the reported cases were included as covariates. This step was performed using SaTScan software (version 10.2.5, National Cancer Institute, Bethesda, MD, USA).

All the cartographic bases used for constructing the thematic maps were obtained from the official and publicly available database of the IBGE (https://www.ibge.gov.br/geociencias/organizacao-do-territorio/malhas-territoriais.html). These shapefiles are open-access and released under public domain. All maps presented in this study were produced by the authors using these public datasets.

## Ethical statement

This study utilized only publicly available secondary data in aggregate form, with no potential for individual identification. Thus, the study did not require informed consent or review by an Ethics Committee.

# Results

## Descriptive analysis

Between 2010 and 2023, 1,042,692 new TB cases were reported in Brazil, of which 896,062 were PTB (S2 Fig in S1 File). Overall, 620,704 cases occurred in men (69.3%), yielding a male-to-female ratio of 2.25:1. The median age was 37 years (interquartile range [IQR], 26–52), and 221,758 cases were recorded in the 20–29-year age group (24.7%). In addition, 538,706 cases (60.2%) occurred among individuals who self-identified as mixed race (*pardo*) or Black, and 652,423 cases (72.8%) were bacteriologically confirmed. Additional demographic and clinical characteristics of PTB cases are presented in S1 Table in S1 File.

## Temporal patterns and trend

Fig 1 shows the monthly counts of PTB cases, which display consistent seasonal variations over the years, with peaks occurring predominantly in March and August and sharper declines in February, June, and December. A downward trend emerged in late 2019 and intensified in March 2020, coinciding with the onset of the COVID-19 pandemic's impact. Case numbers recovered gradually from the last quarter of 2020, reaching the average 2019 level only in October 2021. This pattern was similar across all five regions of the country (S3 Fig in S1 File).

The mean PTB incidence rate during the study period was 30.3 cases per 100,000 inhabitants-years (ranging from 31.2 in 2010 to 33.6 per 100,000 in 2023). Overall, the long-term trend was stable (AAPC = 0.87; 95% CI: −0.04 to 1.34). However, a significant upward trend was observed between 2021 and 2023 (APC = 7.34; 95% CI: 0.58 to 11.79), consistent with the temporal pattern in Fig 1. Across the entire period, increasing trends were documented among men (AAPC = 1.34; 95% CI: 0.81 to 1.78) and in the 20–29-year age group (AAPC = 2.22; 95% CI: 1.11 to 3.32) (Table 1).

The North Region registered the highest incidence (43.7 cases per 100,000) and was the only region to show an increase trend (AAPC = 2.24; 95% CI: 1.77 to 2.77). Nevertheless, more than two-thirds of all reported cases were concentrated in the Southeast (45.1%; n = 403,730) and Northeast (26.5%; n = 237,448) regions (S1 Table in S1 File).

At the state level, the highest incidence rates were observed in Amazonas (64.4 per 100,000), Rio de Janeiro (55.1 per 100,000), Acre (45.8 per 100,000), Pará (43.9 per 100,000), and Pernambuco (41.7 per 100,000). Together, these five states accounted for almost one-third (32.3%; n = 289,546) of all cases reported during the study period (S4 Fig in S1 File). An increasing temporal trend was detected in 13 states and the Federal District—six in the North, four in the

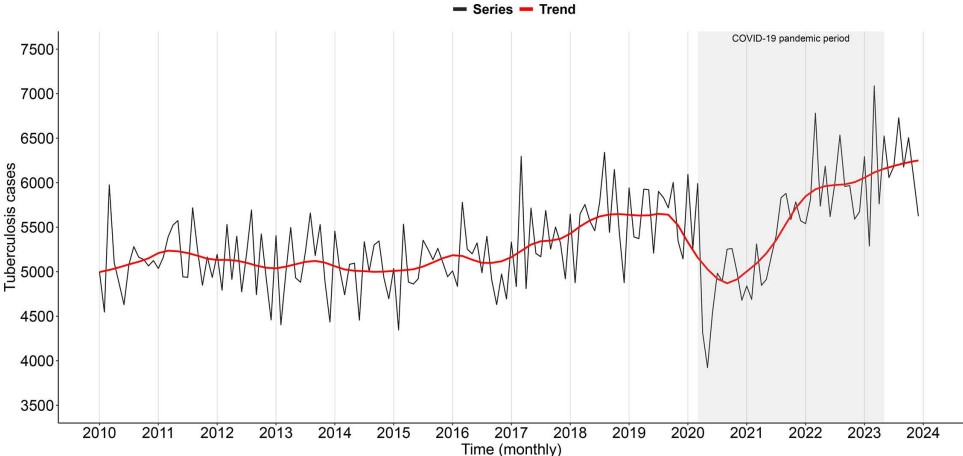

**Fig 1. Monthly time series and temporal trend of tuberculosis cases in Brazil, 2010–2023.** Source: Prepared by the authors.

Northeast, three in the Central-West (two states plus the Federal District), and one in the Southeast. Bahia (Northeast) and Mato Grosso (Central-West) were the only states to exhibit decreasing trends over the period (AAPC = −2.54; 95% CI: −4.04 to −1.11 and AAPC = −2.72; 95% CI: −5.14 to −0.29, respectively). All other states showed stationary trends (Table 1).

## Spatial patterns

Between the initial (2010–2012) and final (2020–2023) study periods, the number of municipalities with PTB incidence rates ≥ 100 cases per 100,000 inhabitants-years increased by 42.8%, rising from 49 to 70. Most of these municipalities were concentrated in the Southeast Region (57.1%, n = 40), mainly in São Paulo (n = 29) and Minas Gerais (n = 10), and in the South Region (15.7%, n = 11), with Rio Grande do Sul accounting for eight of them (Fig 2).

In addition, significant positive spatial autocorrelation was detected for 2010–2023 (Moran's I = 0.233; p = 0.001), and this pattern persisted across all four study subperiods (S2 Table in S1 File). Fig 3 presents the results of the local spatial autocorrelation analysis, highlighting hot and cold spots identified with the Getis-Ord local $G_i^*$ statistic.

In total, 804 municipalities were classified as hotspots between 2010 and 2023, of which 168 (20.9%) persisted throughout every study period; these were located chiefly in the Southeast (49.4%) and North (17.3%) regions. Over time, the number of hotspot municipalities decreased in most regions of the country, except in the North, where the proportion rose from 11.6% (51/438) in 2010–2012 to 20.9% (84/402) in 2020–2023. The largest increases were observed in the states of Roraima (200%), Acre (180%), Amazonas (100%), and Pará (35.7%). Growth in the number of hotspots was also noted in Pernambuco (Northeast), Minas Gerais (Southeast), and Rio Grande do Sul (South). Notably, none of the study periods showed coldspots in Amazonas, Roraima, or Rio de Janeiro, whereas the Federal District exhibited no hotspots at any time (S3 Table in S1 File).

## Spatiotemporal clustering analysis

The space–time scan statistic identified one most likely cluster and 27 secondary high-risk clusters for PTB transmission in Brazil (Fig 4 and Table 2). Together, these clusters encompassed 379 municipalities, fewer than half of all hotspot municipalities detected (47.1%, 379/804), and were located mainly in the North (42%, n = 159) and Southeast (26.4%, n = 100) regions.

**Table 1. Temporal trend analysis of pulmonary tuberculosis cases according to epidemiological variables in Brazil, 2010-2023.**

| Variables | Rate[a] | | | Segmented period | | | | Entire period | | |
|---|---|---|---|---|---|---|---|---|---|---|
| | 2010 | 2023 | 2010-2023 | Period | APC | 95% CI | Trend | AAPC | 95% CI | Trend |
| Brazil | 31.2 | 33.6 | 30.3 | 2010–2021 | −0.32 | −1.90 to 0.20 | Stable | 0.87 | −0.04 to 1.34 | Stable |
| | | | | 2021–2023 | 7.34[b] | 0.58 to 11.79 | Increasing | | | |
| **Sex** | | | | | | | | | | |
| Male | 42.9 | 49.8 | 43.0 | 2010–2013 | −2.42[b] | −6.03 to −0.20 | Decreasing | 1.34[b] | 0.81 to 1.78 | Increasing |
| | | | | 2013–2018 | 2.22[b] | 1.41 to 4.75 | Increasing | | | |
| | | | | 2018–2021 | −3.94[b] | −5.67 to −1.78 | Decreasing | | | |
| | | | | 2021–2023 | 13.71[b] | 9.31 to 17.23 | Increasing | | | |
| Female | 20.1 | 18.9 | 18.2 | 2010–2021 | −1.33[b] | −3.50 to −0.84 | Descending | 0.01 | −0.87 to 0.38 | Stable |
| | | | | 2021–2023 | 7.67[b] | 0.40 to 10.84 | Increasing | | | |
| **Age group (years)** | | | | | | | | | | |
| 0–9 | 3.6 | 5.5 | 3.7 | 2010–2021 | −0.98 | −9.88 to 1.57 | Stable | 3.44 | −0.21 to 5.38 | Stable |
| | | | | 2021–2023 | 31.60[b] | 2.11 to 49.76 | Increasing | | | |
| 10–19 | 12.7 | 15.9 | 14.0 | 2010–2023 | 0.62 | −0.30 to 1.56 | Stable | 0.62 | −0.30 to 1.56 | Stable |
| 20–29 | 42.6 | 49.8 | 44.1 | 2010–2023 | 2.22[b] | 1.11 to 3.32 | Increasing | 2.22[b] | 1.11 to 3.32 | Increasing |
| 30–39 | 42.2 | 43.6 | 38.1 | 2010–2020 | −1.64[b] | −2.55 to −1.02 | Decreasing | 0.29 | −0.44 to 0.97 | Stable |
| | | | | 2020–2023 | 6.99[b] | 2.36 to 14.79 | Increasing | | | |
| 40–49 | 45.6 | 40.6 | 37.9 | 2010–2013 | 8.67[b] | −18.07 to −0.88 | Descending | −1.88[b] | −3.03 to −0.16 | Decreasing |
| | | | | 2013–2023 | 0.25 | −2.11 to 6.89 | Stable | | | |
| 50–59 | 38.1 | 41.0 | 39.7 | 2010–2023 | −1.20[b] | −2.03 to −0.41 | Decreasing | −1.20 [b] | −2.03 to −0.41 | Decreasing |
| 60–69 | 41.2 | 32.1 | 35.9 | 2010–2023 | −1.52[b] | −2.41 to −0.65 | Decreasing | −1.52[b] | −2.41 to −0.65 | Decreasing |
| 70–79 | 41.1 | 35.9 | 34.1 | 2010–2020 | −3.16[b] | −5.15 to −2.20 | Decreasing | −1.00[b] | −2.18 to −0.08 | Decreasing |
| | | | | 2020–2023 | 6.51[b] | 0.12 to 16.36 | Increasing | | | |
| ≥80 | 35.0 | 32.5 | 29.4 | 2010–2023 | −1.03 | −2.30 to 0.15 | Stable | −1.03 | −2.30 to 0.15 | Stable |
| **Region/State** | | | | | | | | | | |
| North | 39.8 | 53.3 | 43.7 | 2010–2015 | −0.07 | −3.89 to 1.50 | Stable | 2.24[b] | 1.77 to 2.77 | Increasing |
| | | | | 2015–2023 | 3.72[b] | 2.90 to 5.52 | Increasing | | | |
| Rondônia | 25.5 | 34.4 | 29.1 | 2010–2012 | 8.74[b] | 1.78 to 13.75 | Increasing | 1.90[b] | 0.91 to 2.59 | Increasing |
| | | | | 2012–2021 | −1.45[b] | −3.92 to −0.86 | Decreasing | | | |
| | | | | 2021–2023 | 10.97[b] | 2.41 to 15.6 | Increasing | | | |
| Acre | 36.4 | 53.6 | 45.8 | 2010–2016 | 0.47 | −1.88 to 1.78 | Stable | 2.56 | 1.75 to 3.10 | Increasing |
| | | | | 2016–2020 | 8.58[b] | 5.98 to 12.34 | Increasing | | | |
| | | | | 2020–2023 | −0.97 | −7.93 to 2.10 | Stable | | | |
| Amazonas | 57.7 | 79.3 | 64.4 | 2010–2023 | 2.61[b] | 1.78 to 3.44 | Increasing | 2.61[b] | 1.78 to 3.44 | Increasing |
| Roraima | 25.5 | 69.6 | 38.0 | 2010–2015 | −4.39 | −11.7 to 0.52 | Stable | 7.35[b] | 5.85 to 8.81 | Increasing |
| | | | | 2015–2023 | 15.42[b] | 12.46 to 19.39 | Increasing | | | |
| Pará | 41.5 | 51.4 | 43.9 | 2010–2014 | −2.96[b] | −6.52 to −0.79 | Decreasing | 1.56[b] | 1.01 to 2.08 | Increasing |
| | | | | 2014–2023 | 3.64[b] | 2.91 to 4.59 | Increasing | | | |
| Amapá | 25.2 | 42.4 | 30.0 | 2010–2014 | −6.56 | −19.71 to 0.57 | Stable | 3.68[b] | 2.16 to 5.59 | Increasing |
| | | | | 2014–2023 | 8.58[b] | 6.24 to 12.6 | Increasing | | | |
| Tocantins | 12.4 | 12.8 | 11.5 | 2010–2016 | −2.72 | −12.5 to 0.23 | Stable | 0.61 | −0.68 to 1.86 | Stable |
| | | | | 2016–2023 | 3.56[b] | 0.99 to 11.81 | Increasing | | | |
| Northeast | 32.1 | 32.2 | 29.6 | 2010–2015 | −2.67[b] | −4.41 to −1.85 | Decreasing | 0.17 | −0.24 to 0.49 | Stable |
| | | | | 2015–2018 | 2.13[b] | 0.30 to 3.40 | Increasing | | | |
| | | | | 2018–2021 | −2.56[b] | −3.99 to −0.87 | Decreasing | | | |
| | | | | 2021–2023 | 8.99[b] | 5.67 to 11.80 | Increasing | | | |

*(Continued)*

| Variables | Rate[a] | | | Segmented period | | | | Entire period | | |
|---|---|---|---|---|---|---|---|---|---|---|
| | 2010 | 2023 | 2010-2023 | Period | APC | 95% CI | Trend | AAPC | 95% CI | Trend |
| Maranhão | 29.2 | 35.7 | 28.7 | 2010–2014 | −4.71[b] | −9.92 to −1.54 | Decreasing | 0.99[b] | 0.30 to 1.76 | Increasing |
| | | | | 2014–2023 | 3.63[b] | 2.59 to 5.04 | Increasing | | | |
| Piauí | 21.3 | 21.3 | 18.4 | 2010–2015 | −7.55[b] | −17.29 to −3.39 | Decreasing | −1.24 | −2.45 to 0.15 | Stable |
| | | | | 2015–2023 | 2.92[b] | 0.68 to 8.17 | Increasing | | | |
| Ceará | 37.3 | 29.6 | 32.9 | 2010–2023 | −0.79 | 2.61 to −0.99 | Stable | −0.79 | 2.61 to −0.99 | Stable |
| Rio Grande do Norte | 24.7 | 29.3 | 29.0 | 2010–2015 | −1.80[b] | −5.72 to −0.04 | Decreasing | 1.41[b] | 0.39 to 1.92 | Increasing |
| | | | | 2015–2018 | 10.44[b] | 5.10 to 13.50 | Increasing | | | |
| | | | | 2018–2023 | −0.50 | −5.58 to 1.32 | Stable | | | |
| Paraíba | 23.6 | 27.6 | 23.7 | 2010–2023 | 0.54 | −1.33 to 2.40 | Stable | 0.54 | −1.33 to 2.40 | Stable |
| Pernambuco | 41.3 | 49.1 | 41.7 | 2010–2021 | −0.39 | −2.03 to 0.26 | Stable | 1.28[b] | 0.12 to 1.83 | Increasing |
| | | | | 2021–2023 | 11.08[b] | 1.60 to 15.50 | Increasing | | | |
| Alagoas | 28.8 | 28.1 | 27.0 | 2010–2018 | −0.87 | −1.66 to 1.92 | Stable | −0.20 | −1.16 to 0.60 | Stable |
| | | | | 2018–2021 | −7.10[b] | −7.10 to −10.01 | Decreasing | | | |
| | | | | 2021–2023 | 14.16[b] | 4.42 to 21.24 | Increasing | | | |
| Sergipe | 20.6 | 36.1 | 28.6 | 2010–2018 | 5.41[b] | 4.77 to 6.47 | Increasing | 4.98[b] | 4.13 to 5.42 | Increasing |
| | | | | 2018–2021 | −1.29 | −3.40 to 2.04 | Stable | | | |
| | | | | 2021–2023 | 13.24[b] | 6.16 to 18.02 | Increasing | | | |
| Bahia | 33.3 | 26.4 | 25.3 | 2010–2023 | −2.54[b] | −4.04 to −1.11 | Decreasing | −2.54[b] | −4.04 to −1.11 | Decreasing |
| Southeast | 32.4 | 35.1 | 31.6 | 2010–2023 | 0.15 | −0.77 to 1.08 | Stable | 0.15 | −0.77 to 1.08 | Stable |
| Minas Gerais | 16.2 | 15.7 | 14.5 | 2010–2015 | −3.72[b] | −6.16 to −2.61 | Decreasing | −0.17 | −0.74 to 0.20 | Stable |
| | | | | 2015–2018 | 1.58 | −1.09 to 3.02 | Stable | | | |
| | | | | 2018–2021 | −3.12[b] | −4.89 to −0.81 | Decreasing | | | |
| | | | | 2021–2023 | 11.40[b] | 6.51 to 14.99 | Increasing | | | |
| Espírito Santo | 31.9 | 34.5 | 27.5 | 2010–2016 | −4.59[b] | −6.95 to −2.81 | Decreasing | 0.35 | −0.35 to 1.04 | Stable |
| | | | | 2016–2023 | 4.80[b] | 3.23 to 6.83 | Increasing | | | |
| Rio de Janeiro | 61.0 | 62.5 | 55.1 | 2010–2014 | −4.82[b] | −10.62 to −1.92 | Decreasing | 0.02 | −1.08 to 0.84 | Stable |
| | | | | 2014–2018 | 5.23[b] | 2.64 to 10.02 | Increasing | | | |
| | | | | 2018–2021 | −7.86[b] | −11.18 to −3.88 | Decreasing | | | |
| | | | | 2021–2023 | 12.83[b] | 3.93 to 20.21 | Increasing | | | |
| São Paulo | 29.4 | 33.9 | 30.4 | 2010–2016 | 2.88[b] | 1.59 to 4.86 | Increasing | 2.07[b] | 1.20 to 2.66 | Increasing |
| | | | | 2016–2021 | −3.96[b] | −8.00 to −2.26 | Decreasing | | | |
| | | | | 2021–2023 | 16.06[b] | 6.90 to 21.66 | Increasing | | | |
| South | 27.0 | 25.8 | 25.1 | 2010–2023 | −0.46 | −1.41 to 0.45 | Stable | −0.46 | −1.41 to 0.45 | Stable |
| Paraná | 19.1 | 17.6 | 16.9 | 2010–2023 | −0.71 | −1.88 to 0.44 | Stable | −0.71 | −1.88 to 0.44 | Stable |
| Santa Catarina | 22.7 | 23.3 | 21.1 | 2010–2021 | −1.95[b] | −5.63 to −0.66 | Decreasing | 0.18 | −1.53 to 1.10 | Stable |
| | | | | 2021–2023 | 12.76 | −0.34 to 20.20 | Stable | | | |
| Rio Grande do Sul | 37.2 | 35.4 | 35.8 | 2010–2023 | 0.02 | −0.86 to 0.88 | Stable | 0.02 | −0.86 to 0.88 | Stable |
| Central−West | 19.4 | 23.1 | 20.2 | 2010–2023 | 0.25 | −0.68 to 1.17 | Stable | 0.25 | −0.68 to 1.17 | Stable |
| Mato Grosso do Sul | 29.2 | 49.9 | 34.2 | 2010–2015 | −2.51 | −12.01 to 1.43 | Stable | 2.83[b] | 1.60 to 4.21 | Increasing |
| | | | | 2015–2023 | 6.31[b] | 4.24 to 11.48 | Increasing | | | |
| Mato Grosso | 34.3 | 29.5 | 32.5 | 2010–2023 | −2.72[b] | −5.14 to −0.29 | Decreasing | −2.72[b] | −5.14 to −0.29 | Decreasing |
| Goiás | 12.6 | 13.5 | 12.6 | 2010–2018 | 0.31 | −0.01 to 1.79 | Stable | 0.89[b] | 0.41 to 1.22 | Increasing |
| | | | | 2018–2021 | −1.93[b] | −3.35 to −0.37 | Decreasing | | | |
| | | | | 2021–2023 | 7.73[b] | 3.32 to 10.61 | Increasing | | | |

**Table 1.** (Continued)

| Variables | Rate[a] | | | Segmented period | | | | Entire period | | |
|---|---|---|---|---|---|---|---|---|---|---|
| | 2010 | 2023 | 2010-2023 | Period | APC | 95% CI | Trend | AAPC | 95% CI | Trend |
| Distrito Federal | 8.2 | 12.3 | 9.7 | 2010–2018 | 2.17[b] | 0.99 to 5.47 | Increasing | 2.58[b] | 1.01 to 3.65 | Increasing |
| | | | | 2018–2021 | −7.00[b] | −11.03 to −1.10 | Decreasing | | | |
| | | | | 2021–2023 | 20.74[b] | 5.53 to 31.21 | Increasing | | | |

*Abbreviations:* APC, annual percentage change; AAPC, average annual percent change; CI, confidence interval.

[a]Incidence rate per 100,000 inhabitants-years.

[b]Statistically significant (*p*-value < 0.05).

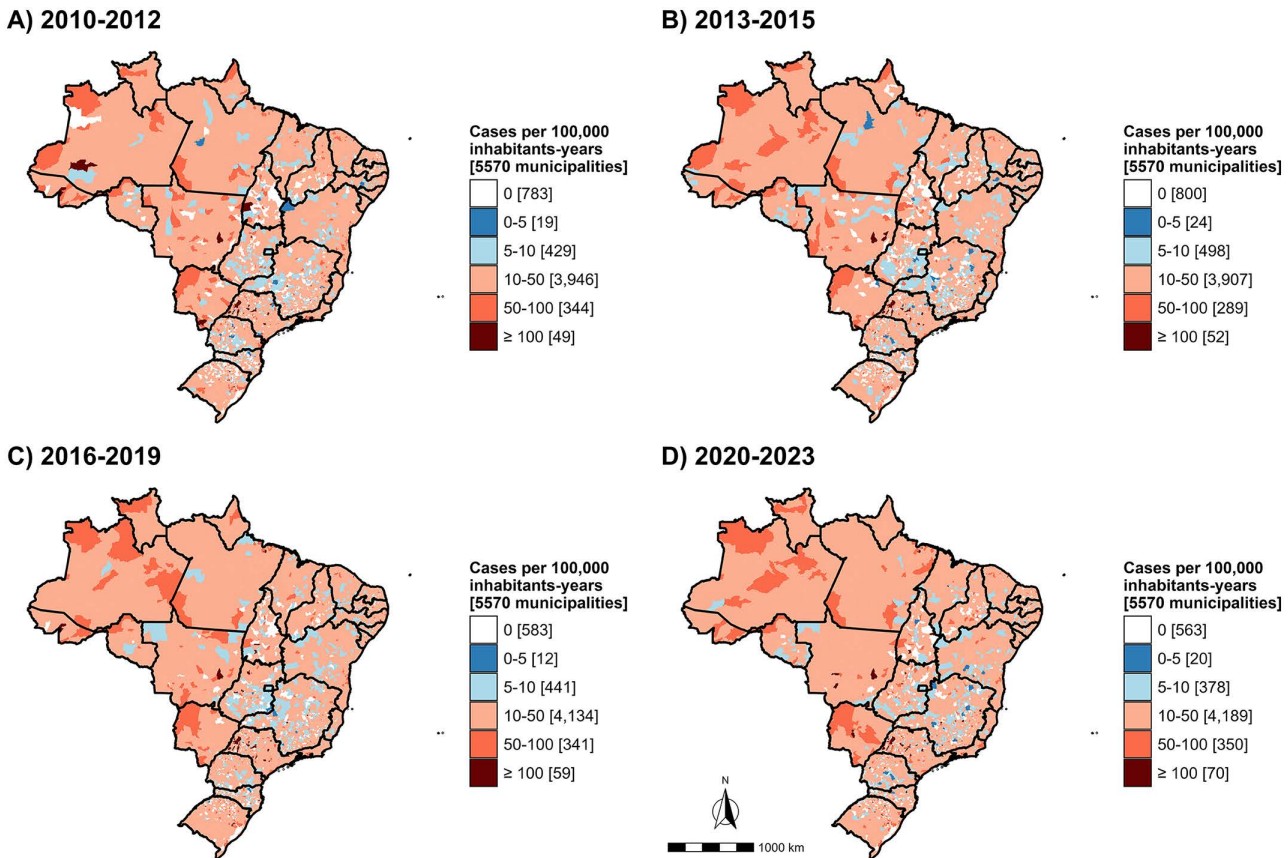

**A) 2010-2012**

**B) 2013-2015**

**C) 2016-2019**

**D) 2020-2023**

**Fig 2. Incidence rate of pulmonary tuberculosis case in Brazil, 2010-2023.** Source: Brazilian Institute of Geography and Statistics (IBGE), municipal territorial mesh (*Malha Municipal*).

The most likely cluster was detected in the Rio de Janeiro Metropolitan Region (RJMR), covering ten municipalities—Nilópolis, Queimados, Belford Roxo, Rio de Janeiro, Duque de Caxias, São João de Meriti, Japeri, Nova Iguaçu, Mesquita, and Magé—between 2017 and 2023. During this period, 56,171 PTB cases (12% of the national total) were reported, yielding an adjusted incidence of 76 per 100,000 inhabitants-years and a RR of 2.51. Notably, the same area, expanded to include five additional RJMR municipalities (Miguel Pereira, Paracambi, Seropédica, Itaguaí, and Engenheiro Paulo de Frontin), was classified as the primary secondary cluster for 2010–2015, with an adjusted incidence of 71.2 per

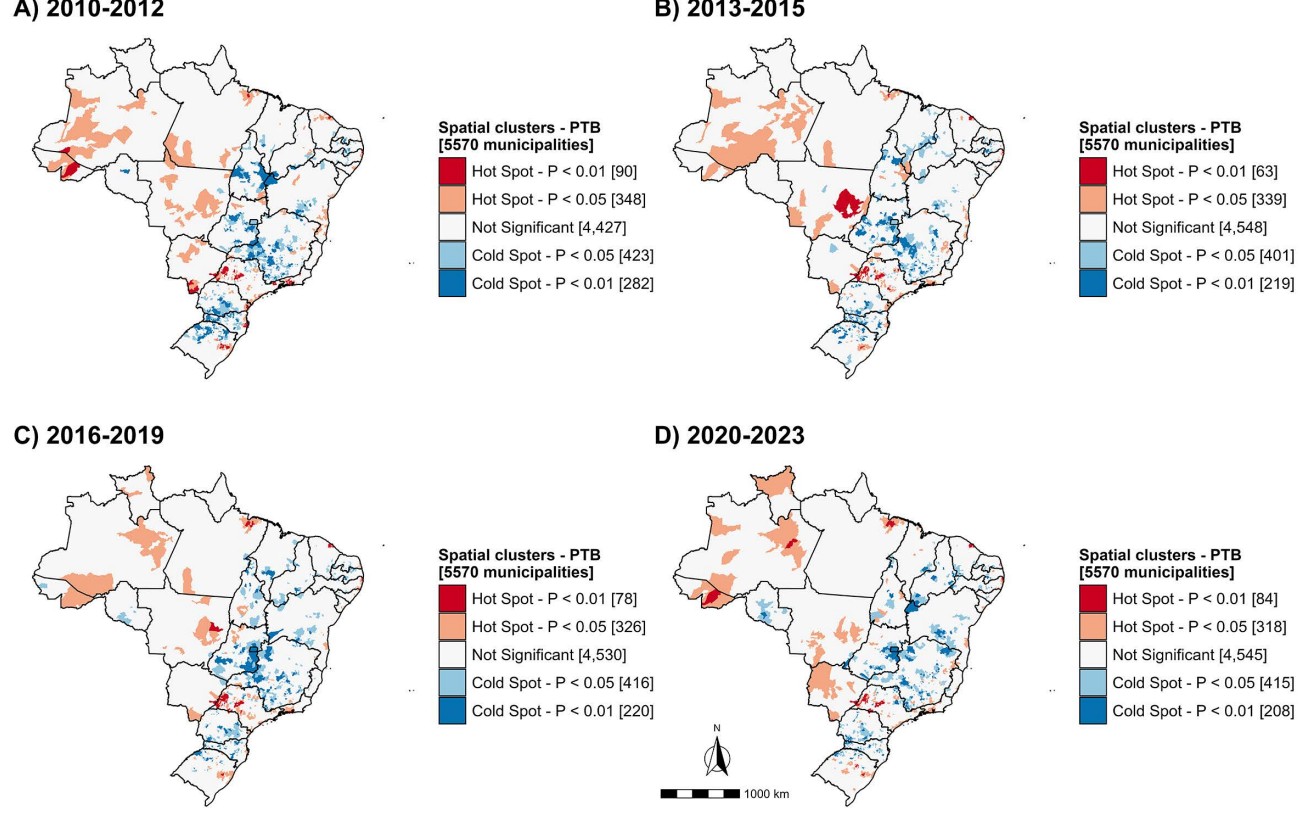

**Fig 3. Spatial patterns of pulmonary tuberculosis cases using Local Getis-Ord $G_i^*$ statistic in Brazil, 2010-2023.** Source: Brazilian Institute of Geography and Statistics (IBGE), municipal territorial mesh (*Malha Municipal*).

100,000 inhabitants-years and an RR of 2.33. Also, in the Southeast Region, twelve clusters were located in the state of São Paulo, spanning 85 municipalities, the largest number for any single state, and accounting for 111,989 PTB cases (12.5% of all cases from 2010 to 2023), with RRs ranging from 1.44 to 7.91.

Clusters 3 and 5 comprised 161 municipalities distributed across six states of the Brazilian Legal Amazon. Their combined RR increased from 1.75 in 2010–2015 to 2.05 in 2017–2023. In the Northeast Region, clusters 4 and 8 were situated in ten municipalities of the Recife Metropolitan Region, Pernambuco, during 2010–2016 (RR = 2.36) and 2018–2022 (RR = 2.23), respectively.

Other secondary clusters included cluster 26 in the São Luís Metropolitan Region (RR = 1.51) in 2015; cluster 14 in the Fortaleza Metropolitan Region (RR = 1.74) for 2010–2013; cluster 12 in the Salvador Metropolitan Region (RR = 1.72) for 2010–2016; cluster 19 in southern Bahia (RR = 1.61) for 2016–2022; and cluster 28 in extreme southern Bahia (RR = 1.71) for 2014–2016.

In the South Region, clusters 6 and 11, both within the Porto Alegre Metropolitan Region, covered 25 municipalities during 2010–2016 (RR = 2.27) and 2018–2022 (RR = 1.95), respectively. In the Central-West Region, clusters 15 and 20, both in Mato Grosso do Sul, spanned 37 municipalities during 2018–2023 (RR = 1.45) and 2010–2016 (RR = 2.38), respectively.

Overall, 212 municipalities remained high-risk clusters for nearly the entire study period and were thus classified as high-priority areas. These municipalities were concentrated mainly in Pará (68), Amazonas (55), São Paulo (36), Amapá

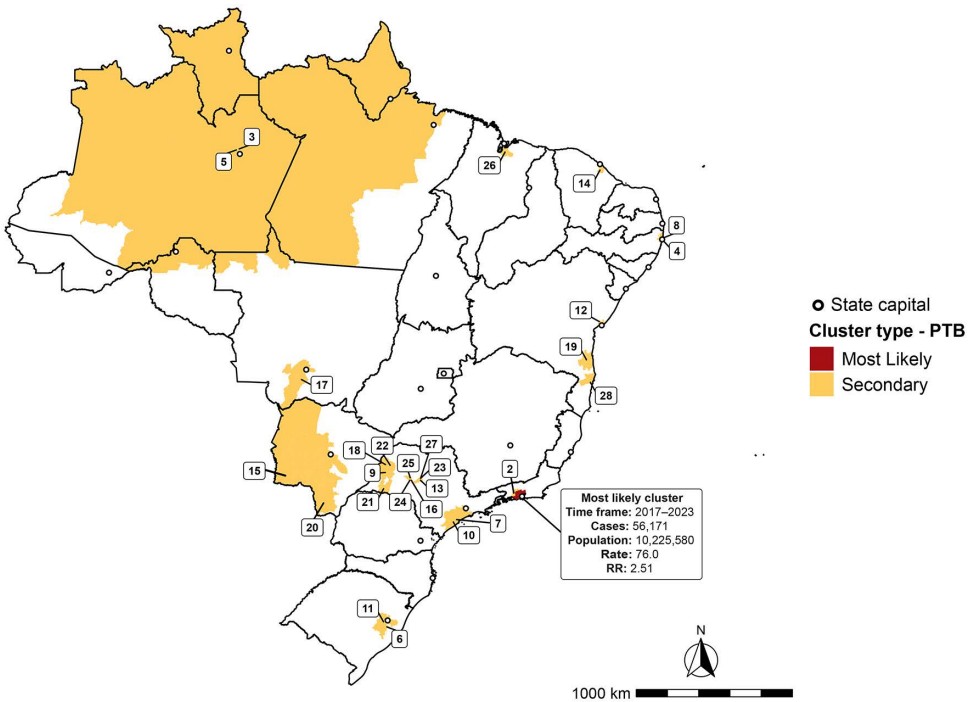

**Fig 4. Spatiotemporal risk clusters for the occurrence of pulmonary tuberculosis in Brazil, 2010–2023.** Source: Brazilian Institute of Geography and Statistics (IBGE), municipal territorial mesh (*Malha Municipal*).

(16), Roraima (15), Rio de Janeiro (10), Rondônia (5), Mato Grosso do Sul (5), and Mato Grosso (2) (Fig 5). The complete list is provided in S4 Table in S1 File.

## Analysis of spatial variation in temporal trends

The SVTT analysis indicated an average annual increase of 0.122% in the PTB incidence rate between 2010 and 2023, reflecting near-stationary growth, consistent with the overall trend shown in Table 1. The scan further identified one most likely cluster and 21 secondary clusters (Fig 6 and S5 Table in S1 File).

The primary cluster included 493 municipalities located across all states of the Legal Amazon. This cluster exhibited an ITT of 1.99% growth per year, whereas the OTT showed a slight decline of −0.20% per year, with a RR of 1.51. The primary secondary cluster was detected in the state of Rio Grande do Norte and included the municipalities of Vila Flor, Senador Georgino Avelino, Nísia Floresta, Arês, and Tibau do Sul. In this cluster, the ITT was 19.35% per year, while the OTT indicated a modest increase of 0.10% per year, with an RR of 3.75. Fig 6 presents the remaining secondary clusters identified in the study, all of which showed increasing trends in both ITT and OTT and were distributed across the various regions of Brazil.

## Discussion

Our study identified critical areas of PTB distribution in Brazil at the municipal level and examined their temporal evolution from 2010 to 2023. We found consistent seasonal variations in monthly case counts throughout the study period. Peaks occurred chiefly in March, followed by August and, to a lesser extent, October, months that coincide with increased demand for health services after vacations, recesses, or surveillance campaigns. The sharpest declines were recorded

**Table 2. Spatiotemporal clusters of pulmonary tuberculosis incidence in Brazil, 2010-2023.**

| Cluster | Time frame | Center/ radius (km) | State | Number of municipalities | Observed | Expected | Rate[a] | RR | LLR | p-value |
|---|---|---|---|---|---|---|---|---|---|---|
| 1 | 2017–2023 | Rio de Janeiro/ 32.17 | Rio de Janeiro | 10 | 56171 | 23225.7 | 76.0 | 2.51 | 17291.4 | <0.001 |
| 2 | 2010–2015 | Rio de Janeiro/ 40.5 | Rio de Janeiro | 15 | 44517 | 19628.4 | 71.2 | 2.33 | 11922.9 | <0.001 |
| 3 | 2017–2023 | Manaus/ 1540.72 | Amapá, Amazonas, Mato Grosso, Pará, Rondônia, Roraima | 161 | 47186 | 23673.9 | 62.6 | 2.05 | 9353.0 | <0.001 |
| 4 | 2010–2016 | Recife/ 27.9 | Pernambuco | 10 | 17481 | 7484.5 | 73.4 | 2.36 | 4888.7 | <0.001 |
| 5 | 2010–2015 | Manaus/ 1540.72 | Amapá, Amazonas, Mato Grosso, Pará, Rondônia, Roraima | 161 | 30345 | 17592.5 | 54.2 | 1.75 | 3883.4 | <0.001 |
| 6 | 2010–2016 | Porto Alegre/ 64.85 | Rio Grande do Sul | 22 | 14416 | 6405.3 | 70.7 | 2.27 | 3719.9 | <0.001 |
| 7 | 2017–2023 | Peruíbe/ 86.9 | São Paulo | 36 | 59826 | 42469.0 | 44.3 | 1.44 | 3321.0 | <0.001 |
| 8 | 2018–2022 | Recife/ 27.9 | Pernambuco | 10 | 12416 | 5600.7 | 69.6 | 2.23 | 3095.2 | <0.001 |
| 9 | 2012–2018 | Presidente Prudente/ 48.69 | São Paulo | 25 | 2928 | 703.4 | 130.8 | 4.17 | 1953.8 | <0.001 |
| 10 | 2010–2015 | Peruíbe/ 86.9 | São Paulo | 36 | 46217 | 34572.0 | 42.0 | 1.36 | 1851.0 | <0.001 |
| 11 | 2018–2022 | Porto Alegre/ 54.53 | Rio Grande do Sul | 23 | 9966 | 5140.3 | 60.9 | 1.95 | 1785.5 | <0.001 |
| 12 | 2010–2016 | Salvador/ 20.32 | Bahia | 5 | 11553 | 6747.3 | 53.8 | 1.72 | 1420.5 | <0.001 |
| 13 | 2012–2018 | Bariri/ 15.76 | São Paulo | 4 | 760 | 96.2 | 248.2 | 7.91 | 907.4 | <0.001 |
| 14 | 2010–2013 | Itaitinga/ 23.72 | Ceará | 9 | 6592 | 3799.8 | 54.5 | 1.74 | 843.8 | <0.001 |
| 15 | 2018–2023 | Porto Murtinho/ 328.47 | Mato Grosso do Sul | 33 | 5279 | 3650.1 | 45.4 | 1.45 | 320.5 | <0.001 |
| 16 | 2013–2019 | Guaimbê/ 26.55 | São Paulo | 4 | 345 | 62.9 | 172.4 | 5.49 | 305.3 | <0.001 |
| 17 | 2016–2022 | Nossa Sra do Livramento/ 115.61 | Mato Grosso | 7 | 3373 | 2207.7 | 48.0 | 1.53 | 265.1 | <0.001 |
| 18 | 2020–2022 | Andradina/ 46.46 | São Paulo | 15 | 653 | 242.6 | 84.6 | 2.69 | 236.2 | <0.001 |
| 19 | 2016–2022 | São José da Vitória/ 45.72 | Bahia | 14 | 1991 | 1239.4 | 50.5 | 1.61 | 192.5 | <0.001 |
| 20 | 2010–2016 | Dourados/ 73.12 | Mato Grosso do Sul | 9 | 667 | 280.1 | 74.8 | 2.38 | 191.9 | <0.001 |
| 21 | 2012–2018 | Presidente Prudente/ 37.75 | São Paulo | 12 | 709 | 322.6 | 69.0 | 2.20 | 172.0 | <0.001 |
| 22 | 2010–2010 | Araçatuba/ 12.77 | São Paulo | 3 | 151 | 21.6 | 220.1 | 7.01 | 164.5 | <0.001 |
| 23 | 2020–2022 | Jaú/ 15.76 | São Paulo | 4 | 179 | 41.4 | 135.9 | 4.33 | 124.6 | <0.001 |
| 24 | 2010–2011 | Lins/ 26.55 | São Paulo | 4 | 84 | 17.4 | 151.3 | 4.81 | 65.5 | <0.001 |
| 25 | 2021–2022 | Lins/ 26.55 | São Paulo | 4 | 78 | 18.1 | 135.5 | 4.31 | 54.1 | <0.001 |

*(Continued)*

**Table 2.** (Continued)

| Cluster | Time frame | Center/ radius (km) | State | Number of municipalities | Observed | Expected | Rate[a] | RR | LLR | p-value |
|---------|-----------|---------------------|-------|--------------------------|----------|----------|---------|-----|------|---------|
| 26 | 2015–2015 | São Luís/ 43.16 | Maranhão | 8 | 633 | 420.0 | 47.3 | 1.51 | 46.7 | <0.001 |
| 27 | 2010–2010 | Bariri/ 15.76 | São Paulo | 4 | 59 | 13.3 | 139.0 | 4.43 | 42.1 | <0.001 |
| 28 | 2014–2016 | Santa Cruz Cabrália/ 43.92 | Bahia | 3 | 245 | 143.0 | 53.8 | 1.71 | 29.9 | <0.001 |

*Abbreviations:* km, kilometers; LLR, log-likelihood ratios; RR, relative risk.

[a] Incidence rate adjusted by sex and age group per 100,000 inhabitants-years.

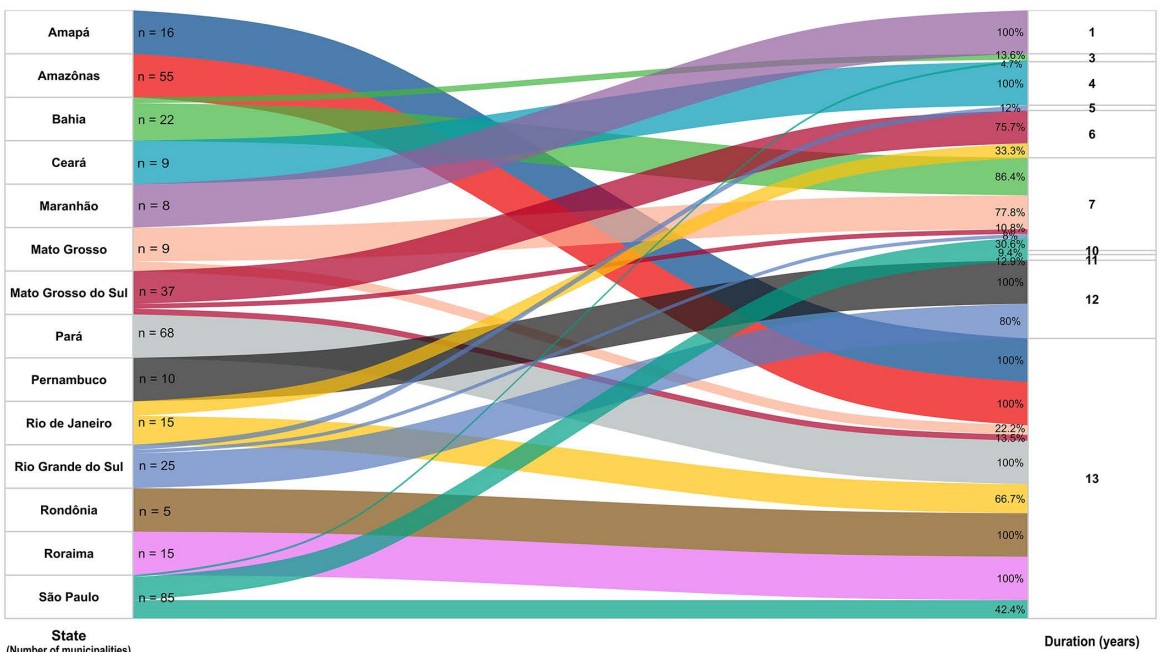

**Fig 5. Flow chart showing the distribution of 379 municipalities by state, classified according to the duration of their status as spatiotemporal risk clusters for pulmonary tuberculosis in Brazil, 2010–2023.** Source: Prepared by the authors.

in February, June, and December, periods traditionally marked by extended holidays, reduced clinic hours, or lower health-service utilization. This regularity suggests seasonality linked to diagnostic opportunity, underscoring the need for surveillance strategies that account for this pattern to optimize TB detection and control [30].

A downturn in case notifications began in the last quarter of 2019 and deepened in March 2020 with the onset of the COVID-19 pandemic. Public fear of visiting health facilities, social-distancing measures, and the diversion of resources to the COVID-19 response markedly reduced the demand for diagnosis and treatment of other diseases, including TB [31]. Partial suspension of TB surveillance and control activities, interruption of screening campaigns, and limited availability of laboratory tests further contributed to underreporting and a temporary weakening of the TB response, explaining the abrupt decline in notifications during 2020 [32].

Consequently, the PTB incidence rate, previously high but relatively stable, fell sharply between 2020 and 2021, returning to pre-pandemic levels only in 2022. Despite this rebound, the results indicate that, although progress has been made

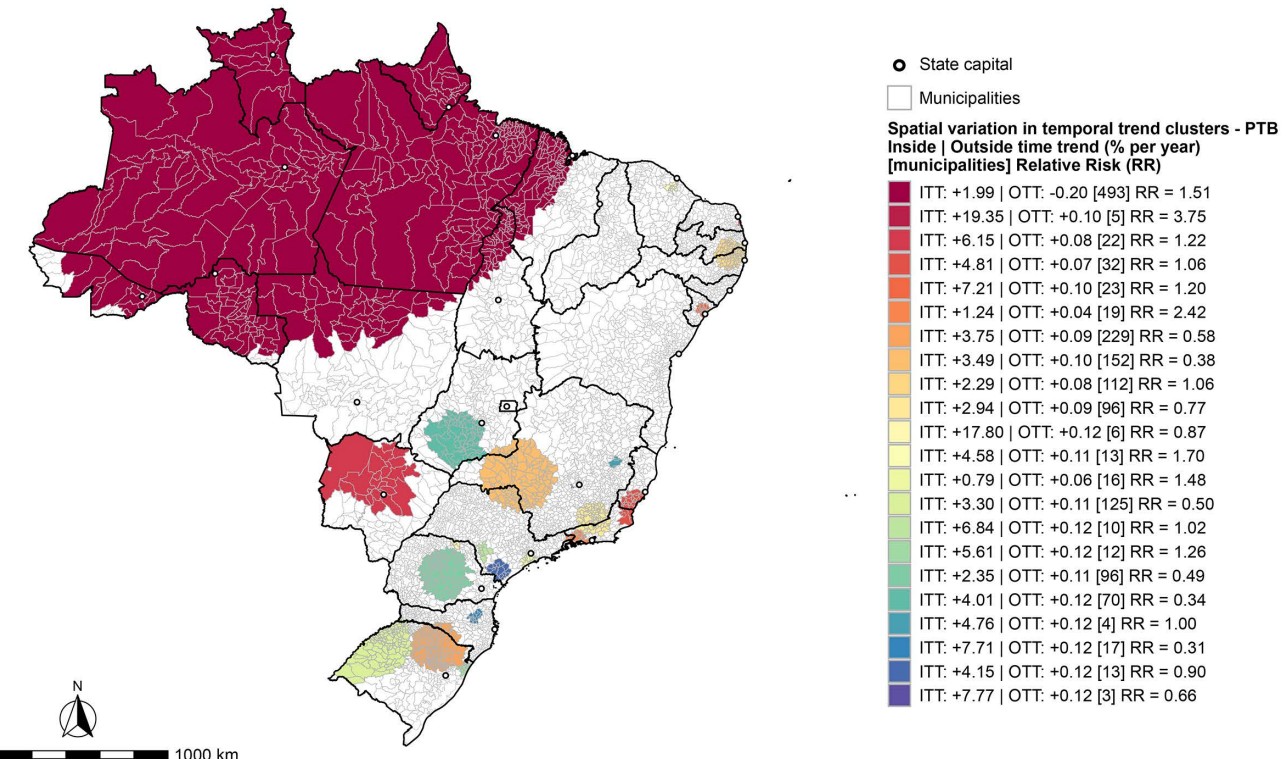

**Fig 6. Spatial variation in temporal trends clusters for the occurrence of pulmonary tuberculosis in Brazil, 2010-2023.** Source: Brazilian Institute of Geography and Statistics (IBGE), municipal territorial mesh (*Malha Municipal*).

in prevention, diagnosis, and treatment, the decline in TB incidence remains insufficient for Brazil to meet the targets set by the World Health Organization (WHO) and by the National Plan to End TB as a Public Health Problem by 2030 [33]. Consistent with these results, a nationwide Bayesian age–period–cohort analysis suggests that PTB incidence in Brazil is unlikely to decline substantially by 2035. Incidence is projected to increase or remain persistently high in metropolitan areas, particularly among men, while non-metropolitan areas are expected to stabilize at levels still far from WHO elimination targets [10].

A recent study partly supports our findings, showing that most TB program indicators in Brazil, including preventive treatment, directly observed therapy, and contact investigation, remain below the 90% coverage targets established by the WHO for 2030 [33]. According to that study, TB incidence in Brazil could reach 38.8 per 100,000 inhabitants by 2025, well above the WHO target of 16.7. To achieve the goals, the authors highlight the need to strengthen program management, address key TB determinants, and intensify integrated, coordinated interventions. Complementing these recommendations, another study projects that public policies aimed at reducing the prison population could lower TB incidence by roughly 16.4% (95% CI: 11.4–23.3) by 2034, representing an important opportunity to reignite progress toward disease elimination [34].

In the same direction, recent evidence shows that scaling up screening and preventive treatment among priority populations in Brazil could prevent approximately 15% (95% CI: 12.8–17.5%) of all TB episodes by 2050 and avert an estimated 1.8 million DALYs (95% CI: 1.2–2.3) over the period. Although expanding these interventions would require up to 62% of Brazil's national TB budget in 2030, costs would gradually decline and remain well below cost-effectiveness thresholds. Notably, the study estimates that every dollar invested would return roughly 51 dollars to

society, reinforcing that preventive interventions are highly cost-effective and yield substantial social, economic, and health benefits [35].

The increasing trend in PTB rates among men and adults aged 20–29 may reflect a combination of factors: greater exposure to high-risk environments, such as precarious workplaces and crowded households, high mobility and economic activity typical of this age group, and barriers to timely health care access that often delay diagnosis and adequate treatment [36,37]. Alcohol and drug use, social vulnerability, and the high TB burden in prisons exacerbate the situation [34,38]. COVID-19-related service disruptions may also have aggravated these inequities and diagnosis delays, partly explaining the upturn observed between 2021 and 2023 [31].

Complementarily, the increasing trends observed in certain states and regions after 2014 may be associated with fiscal austerity policies that gradually weakened public health financing and reduced the responsiveness of the SUS [39]. As noted, these vulnerabilities were further aggravated during the COVID-19 pandemic. Moreover, political instability and the dismantling of health governance structures in recent years, characterized by successive changes in the Ministry of Health leadership and the circulation of anti-scientific rhetoric, further undermined the capacity of the system to meet population needs. This context had a particularly strong impact in historically vulnerable regions such as the Amazon, where health-system collapse became an imminent risk. Recent evidence supports this interpretation, demonstrating that political denialism contributed to increased mortality, declining vaccination coverage, and the weakening of Brazil's National Immunization Program [40], factors that can also directly affect TB control by undermining essential surveillance, prevention, and continuity-of-care efforts.

Given the influence of time on disease distribution, we applied a space–time scan analysis to complement the spatial assessment. This approach identified 15 municipalities in the RJMR as the most likely high-risk cluster for nearly the entire study period, along with 27 secondary clusters throughout the country. The largest secondary cluster comprised 161 municipalities in the Legal Amazon and remained high-risk for 13 years, indicating extensive areas where the PTB transmission chain persists. These areas should therefore be prioritized in upcoming TB control and prevention strategies.

The findings for the RJMR had already been documented in a previous study, that reported an RR of 2.68 in 25 municipalities between 2001 and 2008 [5]. The RJMR houses one of Brazil's largest urban populations and exhibits a combination of structural and social factors that facilitate PTB transmission. High population density, especially in informal settlements such as *favelas*, combined with intense mobility and social inequality creates a favorable environment for disease spread [41]. The region also has historically high TB/HIV coinfection rates [2] and drug-resistant TB rates [42], substantial homeless populations [43], and a large incarcerated population [44], all of which are recognized TB risk factors. These characteristics make TB control in the RJMR a persistent challenge, emphasizing the need for intersectoral, place-based interventions.

Although this study did not directly assess the role of prisons in TB transmission, it is essential to acknowledge that incarceration remains a major structural driver of TB in Brazil [45]. Evidence demonstrated that incarcerated individuals experience extremely high TB incidence, that transmission networks link prisons and surrounding municipalities, and that these patterns are influenced by social inequality and mobility of the incarcerated population [8,46]. Future studies could build upon our findings by incorporating incarceration-related indicators to further clarify the spatial interplay between institutional and community transmission.

The spatial and temporal heterogeneity observed in Brazil parallels patterns reported in other high TB burden countries in Latin America, such as Peru, and Venezuela where socioeconomic vulnerability, overcrowding, and fragmented health systems similarly sustain persistent transmission [1,47]. These findings reinforce that structural inequities, including poverty, housing precarity, incarceration, and migration, are regional determinants of TB, and that spatially targeted, intersectoral interventions are crucial for accelerating progress toward the End TB targets across Latin America [34,48].

Municipalities in the Legal Amazon likewise warrant priority attention in TB control policies, particularly due to persistent adverse socioeconomic indicators that favor transmission [36]. The situation is even more complex in Indigenous

communities, where vulnerability stems not only from poverty but also from broader structural factors such as limited land access, territorial insecurity, and climate-change effects [49,50]. These structural vulnerabilities may also interact with environmental stressors, including the effects of climate variability, which have recently been recognized as emerging determinants of TB dynamics [50].

In addition, informal and unsafe economic activities, such as illegal mining (*garimpos*), deforestation-related labor, and seasonal agricultural work, remain widespread in parts of the Amazon and Central-West regions [51]. These contexts often involve precarious or forced labor conditions, population mobility, and intense exposure to silica dust and smoke from forest fires, all of which may amplify TB vulnerability through both social and biological pathways [52]. Addressing these socioenvironmental determinants is therefore essential for developing integrated TB control strategies adapted to these frontier territories.

Furthermore, the North Region has been the main entry point for Venezuelan migration since 2016, with flows intensifying after 2018 as the economic crisis in Venezuela deepened. This sudden population influx into already fragile areas worsens housing and sanitary conditions, raising PTB transmission risk [53]. Added to this are the logistical challenges typical of the Amazon—vast distances, low population density, and long travel times to health centers—that hamper timely diagnosis and treatment [54,55].

Decentralizing diagnosis and ensuring treatment within primary care, tailored to the sociocultural contexts of Indigenous peoples and refugees, is therefore essential for strengthening TB control nationwide, particularly in the Legal Amazon [36,53,54]. Yet a prior study indicated that the North has the nation's lowest coverage of the GeneXpert MTB/RIF test, a key tool for rapid, accurate TB diagnosis and rifampicin resistance detection [56]. Scarcity of this resource impedes early case detection and timely treatment initiation, especially in remote, vulnerable Amazonian areas.

In São Paulo State, 12 spatiotemporal PTB risk clusters were detected. Among them, clusters 7 and 10, located in the São Paulo and Baixada Santista metropolitan regions, are characterized by high population density, rapid urbanization, pockets of poverty, and pronounced social inequalities [37,57]. These findings reveal ongoing zones of active TB transmission in critical urban settings where precarious housing, overcrowding, intense mobility, and access barriers to regular health care sustain the transmission chain [5–7,37]. São Paulo, Brazil's most populous state and a major internal and international migration hub, also faces complex population dynamics that demand robust, localized responses [58]. Strengthening local surveillance, expanding active case-finding, promoting early diagnosis, and targeting interventions that address social determinants of health are crucial for mitigating the structural inequities underlying TB burden in the state [57].

Additional spatiotemporal PTB clusters were identified in Maranhão, Pernambuco, and Bahia (Northeast) and in Rio Grande do Sul (South), all near their respective metropolitan regions. These results underscore not only the wide spatial and temporal spread of TB but also the need for coordinated public policies across states, especially in densely populated urban areas [10,37].

The SVTT analysis yielded worrisome findings, indicating that without effective interventions in the high-risk areas identified, a substantial rise in PTB incidence is likely in the coming years, particularly in the large cluster encompassing all states of the Legal Amazon. Projected growth in these territories indicates expanding risk areas, including locations not previously flagged as critical. These data point to a worsening epidemiological scenario and call for urgent action by public-health authorities.

## Limitations and strengths

Our study has several limitations. First, the analyses relied on secondary data, which are subject to under-reporting, reporting delays, and inconsistencies, particularly in remote areas. Nonetheless, the quality of TB records in Brazil has improved substantially in recent years [59]. Second, because more disaggregated data, such as residential addresses, were unavailable, all analyses were conducted at the municipal level, which may limit the precise identification of local

transmission foci. However, this level of aggregation allows for a broader, more comprehensive assessment, supporting the identification of priority territories for intervention and reducing the loss of generalizability that can occur with highly disaggregated data. Third, the space–time scan statistic is sensitive to the definition of spatial and temporal scanning windows, which may influence cluster size and boundaries [26]. Furthermore, although sex and age were included as covariates, the method cannot fully capture relevant contextual determinants, such as internal migration flows, population mobility, barriers to health-care access, and local environmental or structural factors, that also influence disease dynamics. Future studies should integrate such contextual variables to provide a more comprehensive, place-based understanding of TB persistence across territories.

Fourth, as an ecological study, the analyses are subject to ecological bias, as associations observed at the municipal level may not reflect individual-level relationships. We also acknowledge the modifiable areal unit problem, whereby results can vary depending on the spatial scale or zoning scheme used, potentially influencing the comparability and interpretation of spatial patterns. In addition, changes in surveillance sensitivity over time may have influenced some of the temporal trends we observed; we could not clearly disentangle the extent to which those trends reflect true changes in disease risk versus fluctuations in case-detection capacity. Finally, the persistence of certain clusters over time may point to entrenched social determinants that are difficult to address, posing an ongoing challenge for TB control in Brazil.

Despite these limitations, the study has important strengths. Foremost is the use of a robust, integrated methodological framework that combines temporal trend analysis, global and local spatial autocorrelation, and space–time scanning. This approach allowed us not only to pinpoint areas with high disease concentration but also to assess their persistence and evolution over time. The broad territorial coverage and inclusion of recent data provide an up-to-date view of PTB distribution in Brazil and highlight critical areas—such as the Rio de Janeiro Metropolitan Region, municipalities in the Legal Amazon, and several in São Paulo State—that warrant priority attention from TB control programs. In addition, the findings are likely applicable to settings with similar socioeconomic and demographic characteristics, particularly in Latin America, thereby informing regional TB-control strategies.

Another strength lies in the practical utility of the results, which can guide the planning and implementation of more effective, locally tailored surveillance and control measures. Finally, the study offers valuable evidence for anticipating future trends, showing that PTB is likely to increase in historically vulnerable areas unless specific, coordinated interventions are adopted.

## Recommendations and public health implications

Urgent enhancements in TB surveillance and diagnostic capacity are critical, especially through the deployment of rapid molecular testing and the decentralization of services in identified high-risk municipalities, remote areas, and Indigenous communities. Given the widespread and entrenched nature of PTB in Brazil, isolated measures will be insufficient. Instead, multi-sectoral strategies must coordinate health care with social assistance, housing, sanitation, and justice services, embedding social-protection policies and the guarantee of basic rights at every level, overall in the population living in vulnerable settings. Finally, sustained, integrated monitoring is essential to track evolving hotspots and gauge the impact of control strategies across Brazil.

## Conclusion

This study provides the most comprehensive spatiotemporal assessment of PTB in Brazil to date, covering municipal-level data from 2010 to 2023. By integrating time-series, spatial autocorrelation, and space–time scan analyses, it identified persistent and expanding high-risk clusters, particularly in RJMR, in the Legal Amazon, and major urban centers.

The study's main contribution lies in revealing how these long-term spatial–temporal dynamics expose structural vulnerabilities that sustain TB transmission, offering actionable evidence for territorial health planning. The identification of 212

municipalities as high-priority for public-health interventions enables more precise prioritization of surveillance, diagnostic expansion, and resource allocation within the national TB control strategy.

These findings underscore the need for structured, regionalized, and intersectoral responses focused on strengthening primary care, decentralizing diagnosis, providing timely and adequate treatment, and reducing social inequalities. Importantly, they reaffirm that TB is not only a biomedical issue but also a social and territorial process shaped by the organization of space and the unequal conditions under which people live. Therefore, without sustained, coordinated efforts, Brazil cannot realistically achieve the WHO's target of eliminating TB as a public-health problem by 2030.

## Supporting information

**S1 File. Supplementary material containing additional figures, tables, and descriptive analyses.**
(DOCX)

## Acknowledgments

We extend our gratitude to all primary care teams within the Unified Health System (SUS) for their commitment and dedication to fight against tuberculosis in Brazil. We also sincerely thank the Brazilian Ministry of Health for providing the data that made this research possible. Additionally, we acknowledge the National Council for Scientific and Technological Development (CNPq – Brazil) for supporting this research through a postdoctoral fellowship awarded to JMNS.

## Author contributions

**Conceptualization:** José Mário Nunes da Silva, Walter Massa Ramalho.

**Data curation:** José Mário Nunes da Silva.

**Formal analysis:** José Mário Nunes da Silva.

**Funding acquisition:** Walter Massa Ramalho.

**Investigation:** José Mário Nunes da Silva.

**Methodology:** José Mário Nunes da Silva, Fredi Alexander Diaz-Quijano, Walter Massa Ramalho.

**Project administration:** José Mário Nunes da Silva, Lívia Teixeira de Souza Maia, Mauro Niskier Sanchez, Ximena Pamela Claudia Díaz Bermúdez, Eduardo de Souza Alves, Walter Massa Ramalho.

**Supervision:** Fredi Alexander Diaz-Quijano, Walter Massa Ramalho.

**Visualization:** José Mário Nunes da Silva.

**Writing – original draft:** José Mário Nunes da Silva.

**Writing – review & editing:** José Mário Nunes da Silva, Fredi Alexander Diaz-Quijano, Lívia Teixeira de Souza Maia, Mauro Niskier Sanchez, Ximena Pamela Claudia Díaz Bermúdez, Eduardo de Souza Alves, Walter Massa Ramalho.

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
