## [Decision Letter · Decision Letter 0]

15 Oct 2025

Dear Dr. da Silva,

After careful consideration, we feel that it has merit but does not fully meet PLOS ONE’s publication criteria as it currently stands. Therefore, we invite you to submit a revised version of the manuscript that addresses the points raised during the review process. Some of them can be discussed in the discussion section and/or as limitations; others will require further analysis. The acceptance of the paper in a second round of review will depend on how you address the feedback from Reviewers 1 and 2. Reviewer 1 recommended major revisions, and although Reviewer 2 recommended acceptance, their comments also reflect a major revision. 

We sincerely appreciate your patience throughout this process.

Warm regards,

Patricia Matsumoto

We look forward to receiving your revised manuscript.

Kind regards,

Patricia Matsumoto, Ph.D.

Academic Editor

PLOS ONE

Journal Requirements:

2. We note that Figures 2, 3, 4 and 6 in your submission contain [map/satellite] images which may be copyrighted. All PLOS content is published under the Creative Commons Attribution License (CC BY 4.0), which means that the manuscript, images, and Supporting Information files will be freely available online, and any third party is permitted to access, download, copy, distribute, and use these materials in any way, even commercially, with proper attribution. For these reasons, we cannot publish previously copyrighted maps or satellite images created using proprietary data, such as Google software (Google Maps, Street View, and Earth). For more information, see our copyright guidelines: http://journals.plos.org/plosone/s/licenses-and-copyright.

  a. You may seek permission from the original copyright holder of Figures 2, 3, 4 and 6 to publish the content specifically under the CC BY 4.0 license. 

Additional Editor Comments:

Comments:

Minors:

- Is the variable source described by the location of probable infection? Is there a chance that this data is recorded by the municipality of residence?

- Although the Discussion briefly mentions the implications of prisons, could you elaborate on how this relates to your results? From the map, there appear to be hotspots in areas with a high concentration of prisons. Please consider adding the locations of prisons in Brazil and examining whether these areas show stronger associations in your analyses. Alternatively, if such an analysis is beyond scope, could you at least cite published studies that document similar patterns?

- Rename the figure names as they appear in the manuscript;

- For the figure 'SVTT_2010a2023', consider including the capitals of each state or at least a point that represent them. In the legend, remove the border of polygons and add some space between them, as in the map, they don't have a border.  

- Change '—' for '–'

Reviewers' comments:

Reviewer's Responses to Questions

**Comments to the Author**

1. Is the manuscript technically sound, and do the data support the conclusions?

Reviewer #1: Yes

Reviewer #2: Yes

2. Has the statistical analysis been performed appropriately and rigorously?

Reviewer #1: Yes

Reviewer #2: Yes

3. Have the authors made all data underlying the findings in their manuscript fully available?

Reviewer #1: Yes

Reviewer #2: Yes

4. Is the manuscript presented in an intelligible fashion and written in standard English?

Reviewer #1: Yes

Reviewer #2: Yes

Reviewer #1: I would like to begin by congratulating the authors for the development of this study, which addresses in a timely and rigorous manner the spatiotemporal distribution of pulmonary tuberculosis in Brazil, with a focus on identifying priority areas for surveillance and intervention. This is a highly relevant topic for public health, especially in a country with significant territorial heterogeneity such as Brazil. The study aims to analyze the spatiotemporal distribution of pulmonary tuberculosis incidence rates in Brazil between 2001 and 2020 and to identify priority areas for disease surveillance and control. The following comments and suggestions are intended to contribute to the improvement of the manuscript.

1. Abstract

I recommend avoiding the use of abbreviations in the abstract. The journal advises minimizing abbreviations whenever possible.

I suggest making the conclusion more concise and highlighting the practical implications of the findings, such as the prioritization of specific areas for intervention.

2. Introduction

I recommend strengthening the theoretical background regarding the social and structural inequalities that influence the distribution of TB in Brazil, particularly in the territories identified as priorities.

3. Materials and Methods

I suggest reorganizing this section into clearer subsections to facilitate readability (e.g., “Unit of analysis,” “Data sources,” “Statistical analysis,” “Spatial analysis”).

I suggest justifying and referencing the exclusion of clusters with ≤2 municipalities, as the rationale presented (random fluctuation) lacks an explicit methodological foundation.

I recommend detailing the procedure used for the proportional redistribution of missing data (sex, age, municipality), including the statistical basis and potential limitations of this method.

I suggest including the limitations of the space-time scan statistic (STSS), particularly regarding its sensitivity to the spatial window boundaries and the lack of contextual variables.

4. Results

I recommend including a quantitative synthesis of the national-level results, such as the overall mean and annual trend of TB incidence in Brazil.

5. Discussion

I suggest condensing the discussion section to avoid repetition of the results and to make the text more analytical and less descriptive.

I recommend explicitly addressing the methodological limitations of the study, including ecological bias and the modifiable areal unit problem (MAUP).

I suggest expanding the discussion on the international and structural implications of the findings, including comparisons with other high TB burden countries in Latin America.

I recommend discussing the limitations of the temporal analysis in the context of the COVID-19 pandemic, considering the possible influence of changes in surveillance and reporting systems during that period.

6. Conclusion

I suggest rewriting the conclusion to avoid repeating content from the abstract and discussion, and to clearly state the original scientific contribution of the study.

I recommend highlighting the main novel findings with direct implications for TB surveillance and territorial planning in Brazil.

7. References

I suggest reviewing the formatting of the references to ensure full compliance with the style required by PLOS ONE, including the presence of DOIs and other mandatory elements.

Reviewer #2: The article “Space-time clustering and temporal trend analysis of pulmonary tuberculosis in Brazil, 2010-2023” aims to analyze space-time trends of PTB in Brazilian municipalities, mapping areas of greater risk and growth, and collaborating with the public health system through the evidence identified.

The paper's introduction presents a clear exposition of the problem, citing information at a global level and the scale of analysis (Brazil). It identifies the need for a temporal and spatio-temporal study of PTB and monitoring and control interventions and actions—especially with a focus on regionalized actions, due to Brazil's regional diversity.

The methodology is precise and robust, featuring sophisticated statistical analysis that is recommended for temporal and spatio-temporal analysis studies. The use of secondary data from DATASUS and IBGE allows for a comprehensive analysis of the scale of the study presented. Furthermore, the use of these open-access databases is a strong point of the study, although the authors mention it as a limitation because they do not analyze disaggregated data. In fact, disaggregated data can provide other clues and suggestions for the problem of PTB in Brazil, however, the greater the scale of detail, the lower the level of generalization for creating transmission models and identifying priority areas.

The use of Joinpoint Regression, Moran's I, Getis-Ord Gi* and SaTScan tools provides a comprehensive and methodologically powerful analysis. The integration between statistical information and spatial statistics takes into account a key element not mentioned in the work: the geography of PTB. Health-disease processes occur in geographical space and are therefore also conditioned/determined by the form of social production of this space. Literature on the subject of the Health Geography—much enriched in Brazil—may provide clues to this point that I would like to draw attention to.

It is clear that the authors have an understanding of the subjects covered in the paper, and some methodological choices have been prioritized in this extensive and rigorous work. Some of the shortcomings are listed in the limitations section of the paper, such as: i) under-reporting; ii) inconsistency in the data; iii) use of aggregated data; iv) those relating to technique (spatio-temporal analysis techniques, MAUP, etc.).

Thus, I highlight some limitations/suggestions for the work. In particular, the areas with the highest demographic density, such as the aforementioned favelas in Rio de Janeiro, are proxies for analyzing the incidence of PTB in Brazil - in contrast to the quality of the data produced in remote/rural areas. From this question, which is deeply energized in urbanization debates in urban geography, a question arises: isn't it important to analyze the consistency and completeness of the data? Another question that I think is relevant to think about is whether there are relationships between the urban hierarchies of Brazilian municipalities and the regions of influence of the cities with the incidence of PTB over the time series. Graphs relating these hierarchies to incidence over time could add to a rich debate about how we are constructing Brazilian cities and regions and the impact this has on Brazilian public health—directly affecting the Unified Health System (SUS). Maps relating to this issue of centrality and the search for health services could add to the debate which, despite the exhaustive analysis presented, could be complemented with some social, economic and structural characteristics.

Other types of maps could correlate areas of socio-economic vulnerability with those with a higher incidence of PTB. The relationship in the Amazon region and the Central-West with precarious and slave labor in illegal mining, deforestation, and wildfires could provide clues to social aspects related to a comprehensive analysis of PTB in Brazil. On this subject, we should also pay attention to recent climatic events resulting from the global climate crisis.

With regard to these more political and social issues (although it is well known that health is a social product), in Figure 2 I identified the municipality of Campinápolis (MT) as an area with a high incidence of tuberculosis in all the periods analyzed. In Figure 3, this region of Campinápolis, in the Serra do Roncador, is home to a lot of illegal deforestation, wildfires and indigenous violence against the Xavantes ethnic group. Are these aspects related? Clearly, the aim of this paper was not to problematize this exact issue, but I recommend that you take a closer look at it in a future article and suggest the concept of necropolitics if you wish to delve deeper into these structural aspects.

Another aspect that I think is important to articulate is the analysis of data based on health regions. The authors clearly know and understand that the SUS is a hierarchical, regionalized, and municipalized health system. As such, resources for surveillance, education, and monitoring of PTB (and other diseases) go through these specific regional departments. If the data were also articulated regionally, it could contribute to the final recommendations made in the paper.

One point that we cannot avoid debating is the COVID-19 pandemic. The abrupt drop in the time series in 2020 and 2021 was mentioned by the authors in the paper and provides an interesting and factual view, since Brazil was one of the countries that recorded the most deaths from the disease, and, in places like the Amazon, there were total collapses of the health system. Another unexplored issue is the institutional dismantling of the SUS, which under Bolsonaro's government has seen major institutional attacks and changes of ministers and officials.

As a final addendum, I think it's relevant that the research brought up the debate on the 2030 Agenda and the complicated situation that Brazil currently finds itself in. In the future, the authors could include a more in-depth analysis with typologies of the municipalities that have historically shown a drop in the PTB trend and which are, or are not, in line with the global agenda. A municipality that has shown a consecutive increase in recent years may be within the limit stipulated by the 2030 Agenda, sparking new debates on the policy.

Finally, I congratulate the authors on the work they have presented, with many figures, tables, and a wealth of supplementary material that fully supports the reading and understanding of the work. Studying Brazil in its plurality and regional diversity spread over 5570 municipalities is a major challenge. Combating tuberculosis and other diseases in Brazil requires coordinated action, and, with this in mind, this work offers relevant insights for the study of tuberculosis in Brazil.

Without further ado, I am available for future dialogues to discuss the relationship between health and geography for future work—I believe that this relationship is fundamental for strengthening the SUS.

**Do you want your identity to be public for this peer review?** For information about this choice, including consent withdrawal, please see our Privacy Policy

Reviewer #1: No

Reviewer #2: No

---

## [Author Response · Author response to Decision Letter 1]

24 Nov 2025

Plos One

Title: Space-time clustering and temporal trend analysis of pulmonary tuberculosis in Brazil, 2010–2023

PONE-D-25-27079.R1

Point-by-point response

COMMENTS TO THE AUTHOR:

Reviewer #1: thanks for addressing the comments.

-1. Abstract

I recommend avoiding the use of abbreviations in the abstract. The journal advises minimizing abbreviations whenever possible.

I suggest making the conclusion more concise and highlighting the practical implications of the findings, such as the prioritization of specific areas for intervention.

Response:

We thank the reviewer for this thoughtful suggestion regarding the abbreviations used in the Abstract. We carefully reviewed this section and maintained the three abbreviations to ensure clarity and fluency, as they are widely recognized and frequently used in the tuberculosis literature.

The abbreviation SINAN (Brazil’s Notifiable Diseases Information System) was kept only once to indicate the original Portuguese name of the database and is not repeated elsewhere in the manuscript. The abbreviation PTB (pulmonary tuberculosis) was maintained to improve readability, as the full term is lengthy and recurrent throughout the text, which could make the reading cumbersome. The abbreviation RR (relative risk) appears only once after its first mention, serving mainly to facilitate comprehension without redundancy. Additionally, the Conclusion has been revised to make it more concise and to emphasize the practical implications of our findings, particularly the identification of priority areas for targeted surveillance and intervention, as shown in the updated version above (lines 20¬¬–23, main manuscript):

“The study identified persistent and expanding patterns of PTB in Brazil, revealing specific areas with higher burden and increasing trends that should be prioritized. These findings reinforce the need for regionally tailored surveillance and control strategies to ensure a more effective response to tuberculosis.”

-2. Introduction

I recommend strengthening the theoretical background regarding the social and structural inequalities that influence the distribution of TB in Brazil, particularly in the territories identified as priorities.

Response:

We thank the reviewer for this valuable comment and fully agree that social and structural inequalities play a central role in shaping the distribution of tuberculosis in Brazil. In response, we have strengthened the theoretical background in the Introduction to better contextualize how socioeconomic disparities, housing conditions, access to healthcare, incarceration, and regional inequalities influence TB burden and persistence across Brazilian territories.

Specifically, we added a paragraph highlighting the interplay between poverty, overcrowding, urban vulnerability, and limitations of health and social systems, as well as their contribution to sustaining endemic transmission in priority areas such as metropolitan regions and the Legal Amazon. These revisions aim to provide a clearer conceptual link between the social determinants of health and the spatial heterogeneity of TB identified in our analyses.

The revised Introduction now reads (excerpt), lines 34–46:

“…

In a country as large and diverse as Brazil, TB distribution is highly heterogeneous, reflecting deep social and structural inequalities. The persistence of poverty, housing shortages, overcrowded environments, malnutrition, and limited access to health services contributes to maintaining endemic transmission, particularly in urban peripheries and territories with poor living conditions [5–7]. Incarceration, migration, and informal labor further exacerbate vulnerability, while regional disparities in health-system capacity and social protection reinforce unequal exposure and outcomes [8,9]. These inequalities explain why certain municipalities and regions, such as large metropolitan areas, border zones, and parts of the Legal Amazon, have remained persistent TB hotspots over time [5,6,10]. Understanding these spatial disparities is therefore crucial for advancing toward equitable TB control and aligning public health strategies with the broader social determinants of health.

…”

We believe these additions strengthen the theoretical grounding of the study and align the Introduction more closely with the reviewer’s recommendation.

-3. Materials and Methods

I suggest reorganizing this section into clearer subsections to facilitate readability (e.g., “Unit of analysis,” “Data sources,” “Statistical analysis,” “Spatial analysis”).

Response:

We thank the reviewer for the thoughtful suggestion regarding the organization of the Materials and Methods section. The manuscript is already structured into specific subsections that correspond closely to the topics mentioned, ensuring clarity and logical flow.

The subsection “Study design and setting” encompasses the unit of analysis, providing a broader contextualization that includes the type of study, the analytical unit (municipalities), and the geographic and epidemiological setting in which the analysis was conducted. Restricting this subsection to the unit of analysis alone would narrow its scope and fragment information that is conceptually interdependent.

The subsection “Study population and data sources” addresses the sources of information used, including surveillance data (SINAN-TB) and demographic indicators, while the “Variables” subsection was intentionally kept separate to describe in detail the indicators derived and their analytical relevance, information necessary for understanding the statistical and spatial modeling procedures.

Finally, the “Statistical analysis” section already includes the recommended analytical components, organized into the following subsections for clarity:

• Time series analysis (to assess temporal trends using segmented regression and seasonal patterns),

• Spatial autocorrelation analysis (using global and local Moran’s I to identify spatial dependence and clustering), and

• Space-time scan statistic analysis (to detect clusters of high or low risk based on Poisson models).

We believe that the current structure provides an integrated and accessible presentation of the methods, allowing readers to follow the analytical process step by step while preserving conceptual coherence.

-I suggest justifying and referencing the exclusion of clusters with ≤2 municipalities, as the rationale presented (random fluctuation) lacks an explicit methodological foundation.

Response:

Thank you for your comment. We recognize the importance of providing a stronger methodological foundation for the exclusion of clusters composed of two or fewer municipalities.

In response, we have expanded the rationale in the Methods section to reflect both methodological guidance and established practice. Although our original justification referenced the potential for random fluctuation, this decision is also grounded in recommendations from the official SaTScan user manual (page 18), which advises reporting only clusters with at least two locations and a minimum number of cases, and with p < 0.05 (https://www.satscan.org/techdoc.html). Therefore, this approach aligns with the software’s default parameters and aims to enhance the statistical robustness and epidemiological interpretability of the detected clusters.

Moreover, clusters formed by only one or two spatial units are widely considered less reliable due to their susceptibility to random variability and the modifiable areal unit problem (MAUP). These clusters are more prone to unstable incidence rates and may lead to spurious results. Similar approaches are documented in the literature, for instance, Huang et al. (2009 [doi:10.1198/jasa.2009.ap07613]) explicitly defined a minimum cluster size of two major statistical areas (MSSAs) or 10 subdivisions to avoid overinterpreting potentially spurious clusters of minimal spatial extent. Likewise, Paiva et al. (2022 [doi:10.1371/journal.pone.0247894]), in their spatiotemporal analysis of tuberculosis in Brazil, excluded clusters with one or two municipalities to mitigate statistical instability and improve the interpretability of the results.

Thus, by applying this exclusion criterion, we follow standard methodological practice to reduce statistical noise and prioritize more stable spatial patterns. The manuscript has been updated to reflect this rationale and now includes appropriate references to support this decision (lines 166¬–170).

“To mitigate statistical noise and enhance the interpretability of results, clusters composed of two or fewer municipalities were excluded from the analysis, as recommended [22]. Very small clusters are more susceptible to random variability and to the modifiable areal unit problem (MAUP), which can lead to unstable incidence estimates and potentially spurious patterns or false positives arising from random fluctuation [5,25].”

-I recommend detailing the procedure used for the proportional redistribution of missing data (sex, age, municipality), including the statistical basis and potential limitations of this method.

Response:

Thank you for this helpful suggestion. We have expanded the Methods section to describe in detail the proportional redistribution procedure applied to records with missing sex, age, or municipality of residence, including its statistical assumptions and limitations.

Briefly, for each analytical stratum, the small number of missing values was redistributed in proportion to the observed distribution within the same group: (i) for sex and age, within the same municipality and calendar year; and (ii) for municipality of residence, within the same state and calendar year. This approach is equivalent to single imputation using empirical (observed) category frequencies under a Missing Completely at Random (MCAR) or Missing at Random (MAR) framework. It preserves totals and prevents artificial deflation of incidence rates in strata with minimal missingness.

We also clarify that proportional redistribution does not incorporate imputation uncertainty and could be biased if missingness were Missing Not at Random (MNAR). However, given the extremely low proportion of missing data (sex: 0.008%; age: 0.03%; residence: 0.02%), the uniformity of missingness patterns across strata, and the structured data collection workflow, we find no evidence of an MNAR mechanism.

The following text has been added to the manuscript (lines 82–91), with references to support the methodological approach:

“…

Records with missing sex (n=76; 0.008%), age (n=235; 0.03%), or municipality of residence (n=168; 0.02%) were handled by proportional redistribution: (i) for sex and age, within the same municipality and calendar year; and (ii) for municipality of residence, within the same state and calendar year. This is equivalent to single imputation using empirical frequencies under missing completely at random (MCAR) or missing at random (MAR) assumptions, preserving totals and avoiding artificial rate deflation [16]. Two caveats apply: it does not propagate imputation uncertainty and could be biased if missingness were missing not at random (MNAR) [17]. However, given the extremely low missingness, uniform patterns across strata, and the data-collection workflow, we find no evidence of an MNAR mechanism [18].”

-I suggest including the limitations of the space-time scan statistic (STSS), particularly regarding its sensitivity to the spatial window boundaries and the lack of contextual variables.

Response:

Thank you for this valuable comment. We agree that the space–time scan statistic (STSS) has methodological limitations that should be acknowledged. Specifically, its results can be sensitive to the definition of spatial and temporal scanning windows, which may influence the size and delineation of detected clusters. To mitigate this effect, we followed SaTScan’s recommended approach of using relative spatial and temporal window limits based on the population at risk and the study period, thereby enhancing robustness and comparability across regions.

Moreover, while the STSS is well suited for identifying statistically significant clusters, it does not incorporate contextual or structural determinants that may underlie the observed patterns—such as internal migration flows, population mobility, barriers to health-care access, and socioeconomic disparities. To partially address this limitation, we included sex and age as covariates in the analysis, controlling for key demographic differences that might otherwise confound spatial patterns.

Accordingly, the Strengths and Limitations section was revised as follows (lines 479–486):

“Third, the space–time scan statistic is sensitive to the definition of spatial and temporal scanning windows, which may influence cluster size and boundaries [25]. Furthermore, although sex and age were included as covariates, the method cannot fully capture relevant contextual determinants, such as internal migration flows, population mobility, barriers to health-care access, and local environmental or structural factors, that also influence disease dynamics. Future studies should integrate such contextual variables to provide a more comprehensive, place-based understanding of TB persistence across territories.”

We believe this revision provides a clearer discussion of the limitations and methodological considerations of the STSS, and we hope these changes address the reviewer’s concern.

-4. Results

I recommend including a quantitative synthesis of the national-level results, such as the overall mean and annual trend of TB incidence in Brazil.

Response:

Thank you for this constructive suggestion. We agree that summarizing the national-level results strengthens the clarity of the Results section.

The overall mean incidence rate was already presented in the manuscript under the subsection Temporal patterns and trend (line 209, Table 1). To address the reviewer’s comment and enhance the quantitative synthesis, we have now also included the Average Annual Percent Change (AAPC) to describe the national temporal trend.

The revised text reads as follows (lines 209–211):

“The mean PTB incidence rate during the study period was 30.3 cases per 100,000 inhabitants-years (ranging from 31.2 in 2010 to 33.6 per 100,000 in 2023). Overall, the long-term trend was stable (AAPC = 0.87; 95% CI: –0.04 to 1.34).”

- I suggest condensing the discussion section to avoid repetition of the results and to make the text more analytical and less descriptive.

Response:

Thank you for this valuable suggestion. Following your recommendation, we carefully revised and condensed the Discussion section to reduce redundancy and strengthen its analytical focus. Descriptive passages that repeated results were streamlined, and transitions were refined to improve clarity and flow.

Additionally, as suggested by the Academic Editor, we incorporated a new paragraph discussing the role of prisons in tuberculosis transmission, providing contextual support from the literature and linking it to our findings in high-incidence areas.

We believe these changes make the Discussion more concise, interpretive, and aligned with the journal’s standards, and we hope these revisions address the reviewer’s concern.

- I recommend explicitly addressing the methodological limitations of the study, including ecological bias and the modifiable areal unit problem (MAUP).

Response:

Thank you for this important comment. The manuscript already addresses this issue in the Strengths and Limitations section, where we discuss the modifiable areal unit problem (MAUP). This discussion highlights that spatial results may be influenced by the level of geographic aggregation (e.g., municipal boundaries), which can affect comparability across studies and the interpretation of cluster patterns. However, we have expanded this section to explicitly include ecological bias, clarifying that, as an ecological study, the analyses are based on aggregated data and therefore do not allow for causal inferences at the individual level.

The revised text reads (lines 487–491):

“Fourth, as an ecological study, the analyses are subject to ecological bias, as associations observed at the municipal level may not reflect individ

---

## [Editor Report · Decision Letter 1]

14 Dec 2025

Space-time clustering and temporal trend analysis of pulmonary tuberculosis in Brazil, 2010–2023

PONE-D-25-27079R1

Dear Dr. da Silva,

We’re pleased to inform you that your manuscript has been judged scientifically suitable for publication and will be formally accepted for publication once it meets all outstanding technical requirements.

Kind regards,

Patricia Matsumoto, Ph.D.

Academic Editor

PLOS One

Additional Editor Comments:

Apologies for the delay in finalizing the evaluation of your paper. I want to acknowledge the excellent job the authors have done in revising the manuscript. The paper is now much stronger, and it is clear that you incorporated the reviews with care and attention. I truly appreciate the effort in addressing each comment from the reviewers and myself, as well as the improvements made to the manuscript, figures, and supporting information.

I only have a few minor points for you to consider. These do not affect the merit of the work, but I believe they will help add the final touch to your manuscript. You don't need to answer them, but consider them for next editorial process.

Minor Points:

- Some institutional names are in Portuguese, while others are in English. Would the authors consider standardizing them?

- Line 22: consider specifying “regionally and nationally”, as the study is conducted at a national scale and can help the federal government in decision-making

- Could you provide a list of the 212 high-priority municipalities in the Supplementary Materials? This can serve for decision-making.

- Figure 6: Since you chose to include the municipal boundaries and they are currently in white, please consider changing them to light gray. This will improve visibility both in the Amazon region and in the white areas where boundaries disappear.

- Table S4: instead of repeating the captions, please add something such as “[continued...]”. As currently formatted, it appears to be a separate table.

Thank you for submitting your paper to PLOS ONE. We hope you will consider future submissions.

All the best,

Patricia

---

## [Editor Report · Acceptance letter]

PONE-D-25-27079R1

PLOS One

Dear Dr. da Silva,

I'm pleased to inform you that your manuscript has been deemed suitable for publication in PLOS One. Congratulations! Your manuscript is now being handed over to our production team.

Kind regards,

on behalf of

Dr. Patricia S. S. Matsumoto

Academic Editor

PLOS One